# Formative Assessment to Improve Cancer Screenings in American Indian Men: Native Patient Navigator and mHealth Texting

**DOI:** 10.3390/ijerph19116546

**Published:** 2022-05-27

**Authors:** Ken Batai, Priscilla R. Sanderson, Lori Joshweseoma, Linda Burhansstipanov, Dana Russell, Lloyd Joshweseoma, Chiu-Hsieh Hsu

**Affiliations:** 1Department of Urology, College of Medicine-Tucson, The University of Arizona, Tucson, AZ 85724, USA; 2The University of Arizona Cancer Center, Tucson, AZ 85724, USA; pchhsu@arizona.edu; 3Health Sciences Department, College of Health and Human Services, Northern Arizona University, Flagstaff, AZ 86011, USA; 4Department of Health and Human Services, Hopi Tribe, Kykotsmovi, AZ 86039, USA; ljoshweseoma@hopi.nsn.us (L.J.); lljoshweseoma@hopi.nsn.us (L.J.); 5Native American Cancer Research Corporation, Pine, CO 80470, USA; burhansstipanov@gmail.com; 6HOPI Cancer Support Services, Department of Health and Human Services, Hopi Tribe, Kykotsmovi, AZ 86039, USA; drussell@hopi.nsn.us; 7Department of Epidemiology and Biostatistics, Mel and Enid Zuckerman College of Public Health, The University of Arizona, Tucson, AZ 85724, USA

**Keywords:** American Indian men, mHealth, Native Patient Navigator, Community-Based Participatory Research

## Abstract

Cancer screening rates among American Indian men remain low, without programs specifically designed for men. This paper describes the Community-Based Participatory Research processes and assessment of cancer screening behavior and the appropriateness of the mHealth approach for Hopi men’s promotion of cancer screenings. This Community-Based Participatory Research included a partnership with H.O.P.I. (Hopi Office of Prevention and Intervention) Cancer Support Services and the Hopi Community Advisory Committee. Cellular phone usage was assessed among male participants in a wellness program utilizing text messaging. Community surveys were conducted with Hopi men (50 years of age or older). The survey revealed colorectal cancer screening rate increased from 51% in 2012 to 71% in 2018, while prostate cancer screening rate had not changed (35% in 2012 and 37% in 2018). Past cancer screening was associated with having additional cancer screening. A cellular phone was commonly used by Hopi men, but not for healthcare or wellness. Cellular phone ownership increased odds of prostate cancer screening in the unadjusted model (OR 9.00, 95% CI: 1.11–73.07), but not in the adjusted model. Cellular phones may be applied for health promotion among Hopi men, but use of cellular phones to improve cancer screening participation needs further investigation.

## 1. Introduction

American Indians and Alaska Natives have heavier burdens of chronic diseases, higher death rates from multiple diseases, including cancer, and lower life expectancy than Non-Hispanic Whites or other racial/ethnic groups in the United States (U.S.) [1,2,3,4,5,6,7]. American Indian/Alaska Native men compared to American Indian/Alaska Native women also have higher death rates from heart disease, cancer and chronic liver disease [1]. Cancer health disparities exist with 11% higher overall cancer mortality rate among American Indian/Alaska Native men compared to Non-White Hispanic men. Over the years, Non-White Hispanics’ mortality rates have declined for many cancer types, including two major screen-detectable malignancies among men: colorectal cancer (CRC) and prostate cancer (PCa). However, cancer mortality rates remain high among American Indian men for some cancer types [8,9,10,11,12]. In Indian Health Service Contract Health Service Delivery Areas, American Indian/Alaska Native men have 37% higher CRC and 9% higher PCa mortality rate compared to Non-Hispanic White men [10,11]. American Indian men may have even greater CRC and PCa burden than Non-Hispanic Whites, and reported incidence and mortality rates may be lower than the real burden due to lower cancer screening rate, misclassification of race/ethnicity in registry data, and lack of follow-up after abnormal findings [13,14,15]. In Arizona, PCa, CRC, and kidney cancer are the top three most common cancer types among American Indian men [16]. American Indian men are more likely to be diagnosed with more advanced-stages of PCa than Non-Hispanic White men (17% of American Indians vs. 5% Non-Hispanic Whites diagnosed with distant PCa) [16]. In the general population, lung and breast cancer have a higher mortality rate than other types of cancers, but for American Indian men in Arizona the PCa mortality rate is higher than the mortality rate of other types of cancers [17].

The high observed mortality for screen detectable cancers in American Indians and Alaska Natives may be partly due to low uptake of cancer screenings. The Centers for Disease Control and Prevention (CDC) funded a National Breast and Cervical Cancer Screening Program targeting low-income, uninsured, and underinsured women including American Indian and Alaska Native women [18]. The program has been, and continues to be, successful. American Indian men do not receive similar federal funding support for a cancer screening program like that for American Indian women, and American Indian men have lower screening rates for CRC and PCa than Non-Hispanic White men [19,20,21,22]. The Behavioral Risk Factor Surveillance System (2000–2010) reported evidence for even lower cancer screening rate for American Indian/Alaska Native men in the Southwest region of the U.S. [21]. Overall CRC screening rate was 44.3% for American Indian/Alaska Native men across the nation, but CRC screening rate was 36.6% among American India/Alaskan Native men in the Southwest. Reasons for low screening rates may include stigma and fear related to diagnosis, distrust of medical professionals, lack of knowledge or awareness, masculinity, lack of transportation, and geographic distance to screening facilities [23,24,25,26]. American Indians also have multiple challenges to healthcare (e.g., lack of healthcare coverage, language and culture, poverty, and discrimination) resulting in low healthcare utilization [21,27,28,29,30]. American Indians are less likely to utilize preventive care or attend healthcare provider appointments/follow-ups [29]. However, a few studies investigated factors contributing to low cancer screening rates specifically for American Indian men [22,25], and factors influencing cancer screening behavior may vary in different American Indian tribes.

For American Indian men who have multiple cultural, socioeconomic, and geographic barriers to healthcare, it may be necessary to employ multiple effective methods to increase cancer screening. The use of Native Patient Navigators and community education have been demonstrated to increase cancer care utilization including screening and to shorten the time between an abnormal finding and diagnosis among American Indians [31,32,33]. Use of mobile health (mHealth) technology, such as text messaging, has also been successfully used in interventions for health promotion [34], including efforts to increase cancer screening in American Indians and Alaska Natives [35]. A concerted approach of mHealth and patient navigation may improve cancer screening participation in American Indian men. However, there is limited data available on mHealth technology use for American Indian men. The use of text messaging for preventive care and cancer screening promotions and reminders among American Indian men living on tribal lands has not been well explored.

To develop a program implemented by a male Native Patient Navigator with mHealth text messaging approach to increase cancer screening participation among Hopi men, a formative assessment using a Community-Based Participatory Research approach was conducted in a Hopi Reservation between September 2017 and March 2019. The aims of the current paper are (1) to describe the Community-Based Participatory Research processes and activities, (2) to assess cancer screening behavior among Hopi men, and (3) to assess the appropriateness of Native Patient Navigator with mHealth approach for Hopi men’s promotion of cancer screenings.

## 2. Materials and Methods

### 2.1. Target Population

The Hopi Tribe is one of 22 federally recognized tribes in the State of Arizona in the Southwest region of the U.S. The Hopi Tribe is a sovereign nation located in a rural remote northeastern part of Arizona. The Hopi Reservation encompasses over 1.5 million acres with three mesas surrounded by 12 villages, including villages on the mesas. There are approximately 14,000 persons enrolled in the Hopi Tribe. About half of the population live on the Hopi Reservation, and approximately 900 Hopi men are within the recommended CRC screening age (between 50 and 75 years). The team targeted Hopi men who were within the recommended CRC screening age and resided on the Hopi Reservation. American Indian men who were not Hopi and Hopi men who reside outside of the Hopi Reservation, and who were 49 years or younger were not eligible to participate.

### 2.2. Approval to Conduct Research

This study was conducted through a partnership among H.O.P.I. (Hopi Office of Prevention and Intervention) Cancer Support Services, Northern Arizona University, and the University of Arizona Cancer Center. The Hopi Tribal Council and Northern Arizona University Institutional Review Board (IRB) approved the study. The University of Arizona IRB approved an IRB deferral to Northern Arizona University IRB. There are two Indian Health Service facilities that Hopi community members utilize, Tuba City Regional Health Care Corporation located right outside of the Hopi Reservation, and Hopi Health Care Center located in the Hopi Reservation. The two regional healthcare centers provided a letter of support for the study.

### 2.3. H.O.P.I. Cancer Support Services

The H.O.P.I. Cancer Support Services applies an adapted cultural vehicle to conduct outreach and communicate behavioral risk factors, and to increase preventive care, cancer care and cancer screening for women. The H.O.P.I. Cancer Support Services’ slogan, *Namitunatya*, translates to “taking care of yourself.” The traditional approach of communication and education has been identified in other studies to be successful for healthcare programs [36,37], as American Indian cultural teachings have traditionally been carried out in-person and through the use of oral narrative [38,39]. This method of oral communication for information sharing and education is still the preferred form of communication in hard-to-reach populations, especially among American Indians [40,41]. In 1996, the CDC funded the Hopi Breast and Cervical Cancer Screening Program for women. The H.O.P.I. Cancer Support Services has female case managers utilizing the traditional in-person educational approach with oral narratives. The traditional cultural approach and female case managers have increased the rate of breast cancer screening (have had a mammography within the past two years) from 26% in 1993 to 71% in 2012 among Hopi women aged 40 or older living in the Hopi Reservation [42,43].

To further determine community needs, H.O.P.I. Cancer Support Services and Hopi Department Health and Human Services conducted a community survey in 2012 (2012 Hopi Survey of Cancer and Chronic Disease). This was a collaborative project between Hopi Tribe, Northern Arizona University, and the University of Arizona [42,44]. It was a population-based survey with randomly selected adult Hopi Tribal members from an enrollment list provided from the Hopi Tribe. The 2012 survey revealed an urgent community need to develop a cancer screening program for Hopi men because of continued low cancer screening rates among Hopi men [22].

During the 2016 Hopi Department of Health and Human Services Health Summit led by one of the co-authors, the Hopi community stakeholders identified several unmet needs which included prioritizing Hopi men’s healthcare. The community stakeholders expressed a need to educate men on cancer screening and prevention as well as preventive care. The following year, the Hopi Department of Health and Human Services in collaboration with Northern Arizona University Health Sciences Department and University of Arizona Cancer Center developed a Community-Based Participatory Research proposal and successfully received a two-year pilot funding from U54 Partnership for Native American Cancer Prevention (NACP).

### 2.4. Community Advisory Committee

The Hopi team recruited five community advisory committee (CAC) members who were all Hopi men (age 50 or older) living on the Hopi Reservation. The CAC were active members of Hopi community including one elder who was a former director of Hopi Department of Health and Human Services and one cancer survivor. One of the CAC members was the first Hopi Native Patient Navigator. He took another position as a health educator in Hopi Department of Health and Human Services and then joined CAC. Another CAC member had previous experience in health research conducted in the Hopi reservation [45]. The project team met quarterly with the CAC on the Hopi Reservation from 9:00 am to 3:00 pm. During these meetings, CAC members provided guidance, feedback, and recommendations related to recruitment strategies, study methods, protocols (research and Hopi culture), research materials, role of Native Patient Navigators, and dissemination strategies from a Hopi perspective. The project advisors attended the meetings to provide suggestions for integrating scientific and community perspectives. Representative from two Indian Health Services facilities who were not Hopi community members were invited to each quarterly meeting with the CAC. When the representatives from the Indian Health Service facilities were present, the representatives provided recommendations related to healthcare utilization to the project team. Northern Arizona University undergraduate and graduate research assistants also attended the quarterly meetings to assist with logistical preparation, including meeting notes. The H.O.P.I. Cancer Support Services manager facilitated quarterly meetings.

### 2.5. Native Patient Navigator and Training

To prepare for a patient navigator program, the project team took an initiative to educate the Hopi team about the patient navigator program. The project team hosted a three-day patient navigator training in Phoenix, Arizona. Breast, cervical, and other cancer patient navigators from other tribes (Navajo Nation and Northwest tribes) were invited and attended the training. The Native American Cancer Initiatives, Inc. (Pine, CO, USA), facilitated the three-day patient navigator and motivational interviewing training. In addition to the Native Patient Navigators, training attendees included CAC members, project team, Northern Arizona University students, and the Director of Hopi Department of Human and Health Services. The training curriculum included Cancer 101, cancer screening methods and guidelines, cancer treatments, Native Patient Navigator’s roles and responsibility, and motivational interviewing. The goals of the training were not only to prepare the first Hopi Native Patient Navigator to take on the role as a navigator but also to educate the research team and Hopi community members on the roles and responsibilities of a navigator. While this is not the scope of the current paper, the long-term goal of this project was to develop a cancer screening program for Hopi men run by a Hopi male Patient Navigator, and the appropriateness of the Hopi male Native Patient Navigator to provide support and service to Hopi men who would be accessing cancer screenings and care was assessed as a part of the project.

A sub-recipient grant was issued by Northern Arizona University to the Hopi Tribe to hire a part-time Hopi male Native Patient Navigator. The project team conducted interviews and hired the first male Native Patient Navigator. However, after a month, he resigned to take a full-time job. To reduce the possibility of this occurring again, the Hopi Native Patient Navigator position included additional job responsibilities with the H.O.P.I. Cancer Support Services to create a full-time position with benefits. The second male Native Patient Navigator was an active member of the Hopi community and was bilingual in the Hopi language and English. Fortunately, he participated in the Native Patient Navigator training in Phoenix as a community advisory committee member. With his transferable job skills, he moved the project forward. The Native Patient Navigator completed human subjects training, Collaborative Institutional Training Initiative (CITI) training, National Institutes of Health Conflict of Interest (COI), and the Health Insurance Portability and Accountability Act (HIPAA) and compliance.

### 2.6. Formative Assessment

The formative assessment was conducted between September 2017 and March 2019. First, the Hopi Tribe had a wellness program, called Buddy Challenge, that encouraged tribal members (men and women who were 18 years and older) to increase their physical activities by working together with their assigned partner. In this program, participants had an option to receive text messages that encouraged them to exercise. Buddy Challenge coordinators sent exercise instructions daily via text. A small group of Hopi community members chose this option, and they received the text messages for eight weeks. The wellness program coordinator sent the text messages throughout the day to inform Buddy Challenge participants to do one-to-three-minute exercises, such as push-ups, jumping jacks, and deep knee squats. The project team planned to recruit and interview 10 male Buddy Challenge participants who were 50 years of age or older to assess their cellular phone usage for a wellness program. The interviews with Buddy Challenge participants included multiple choice questions on receiving text messages, issues with cellular phone services, and experiences of using text messaging and smart phone for a wellness program. Information related to cancer screening was not collected. The Hopi Native Patient Navigators from H.O.P.I. Cancer Support Services conducted interviews at the Hopi Veteran’s Center where the Hopi Wellness program is located.

Second, community-wide surveys were conducted at various locations across the Hopi Reservation. The goal was to assess preventive care utilization, cancer screening, percentage of cellular phone use and mobile technology literacy among 200 Hopi men (aged 50–75 years). The Hopi Native Patient Navigator conducted the surveys using the audience response system (ARS). The ARS has been successfully used in a previous study that conducted community surveys in American Indian communities [46]. When the participants checked in, each participant was provided with a “clicker” to answer the survey questions. An ARS receiver connected to a laptop receives signals from clickers to record answers. PowerPoint with ARS software add-in displays questions and polls real-time. The research team demonstrated the ARS to the community advisory committee, and committee members thought the ARS was appropriate for Hopi men, especially to maintain confidentiality of study participants. The research team analyzed the de-identified data.

### 2.7. Enrollment, Consenting, and Data Collection Process

The Hopi team posted recruitment flyers at various locations and sent organizational and employment-based mass emails to advertise the research project. The research team handed out recruitment flyers at health promotion events and at village stores. The research team also visited the homes of potential participants and explained the research objectives and activities. The Hopi Native Patient Navigators from H.O.P.I. Cancer Support Services also obtained a list of Buddy Challenge participants and called each participant. To increase the number of study participants in the community survey, the committee members recommended raffle prizes.

The Hopi team conducted eligibility screening. If eligible, potential participants from the Buddy Challenge Program were scheduled for an interview. All eligible participants were invited to ARS community survey sessions. Men were able to participate in only one community survey session. Before each research activity, the Hopi team described the purpose of the project, and the benefits and risks of participating in the study. Two informed consent documents were given to each participant; two signatures were completed on two consent documents. One of the signed consent forms was given to the participant for their records, and the second signed consent was for the Hopi project records. A demographic form was then given to the Buddy Challenge program participants to complete. To ensure anonymity, the Hopi team added a random number to the demographic form, so the information from the demographic form could not be traced to the participant.

After the consenting process, the Native Patient Navigator read aloud each question. During the survey with ARS, questions and choices for answers were displayed on the PowerPoint Presentation screen, and the Native Patient Navigator clarified any questions that study participants had. The polls were open for about 30 s for the study participants to enter their answers using their assigned clicker. The use of ARS allowed participants to respond anonymously. The study participants were able to see the poll results after each question, making the survey more interactive.

Upon completion of each research activity, an incentive of $25 Walmart gift card was given to thank participants for their time. Raffle prizes attracted many potential survey participants. After all survey sessions were completed, raffle winners received a $400 chainsaw.

### 2.8. Statistical Analysis

Statistical analysis for data from ARS survey was conducted by the university team. The proportion of men who have had PCa and CRC screening and technology literacy (e.g., proportion of men with smart phone and texting experience) were calculated. Multiple logistic regression was used to assess whether demographic characteristics, comorbidities, healthcare utilization, and electronic health literacy were associated with past CRC and PCa screening participation. The details of the analysis methods have been described previously [22,42]. Briefly, logistic regression was performed for each variable to assess association with ever having CRC or PCa screening and to calculate odds ratios (OR) and the associated 95% confidence intervals (CI). The variables with a significant unadjusted OR at a significance level of 5% were included in adjusted analyses. *p*-values less than 0.05 were considered statistically significant in all analyses. All statistical analyses were performed with SAS 9.4.

## 3. Results

Despite the small size of the community and a narrow age range for eligibility, the research team successfully recruited 89 eligible Hopi men for the interviews with Buddy Challenge Program participants or ARS community surveys.

### 3.1. Evaluation of Text Messaging Use among Buddy Challenge Program Participants

To assess cellular phone usage for a wellness program and to gather information on appropriateness of the mHealth approach, six male Buddy Challenge Program participants ages between 33 and 53 years were interviewed. There were only three Hopi men aged 50 years and older who chose the text message option for the Buddy Challenge Program. Due to the small number of older Hopi men in the Buddy Challenge Program, the team lowered the eligibility criteria for age to 30 years and older upon Northern Arizona University IRB approval. The Native Patient Navigator conducted the interviews individually with the participants.

All participants had smart phones. Three out of six reported having a computer at their home. One did not have an email account. All men sent text messages daily. Three had issues with their cellular phone services on Hopi Reservation. Two said they had problems receiving and sending text messages at home. Three preferred receiving email messages for health promotion, while one preferred receiving message by telephone. Only one reported that he preferred receiving health promotion messages by text message. One had no preference.

### 3.2. Community Audience Response System (ARS)

The team, led by the Native Patient Navigator and H.O.P.I. Cancer Support Services, conducted 11 community ARS survey sessions. There were more study participants in the survey sessions during community-based or employee-focused health events (*n* > 15) than the survey sessions at village level community centers where the number of participants ranged from 1 to 11. The proposed recruitment of 200 men was not met. A total of 91 men completed the ARS survey. Although study participants’ eligibility was screened before each survey session, based on the survey answers, only 83 men were eligible (age 50 or older and reside in the Hopi Reservation).

Table 1 shows the characteristics of Hopi men who participated and completed the survey. More than half of Hopi men (78.3%) spoke Hopi as a primary language at home, and 7.2% of the men were cancer survivors. Two common chronic health conditions were diabetes (43.4%) and hypertension (53.0%), but 71.1% answered that they were in good, very good, or excellent health. About half (51.8%) reported that they had had an annual physical exam within the past year. Only 13% of men reported that they had learned about cancer screening from healthcare providers, while 34.2% of men responded that they had learned from H.O.P.I. Cancer Support Services. The use of electronic devices for healthcare and wellness was low. Only 33% of men reported that they had looked for medical information using electronic devices in the past year, and only 19% of men had wellness apps on their smartphone or tablet.

Figure 1 shows trends in cancer screening participation between 2012 (Hopi Survey of Cancer and Chronic Disease) and 2018–2019 (current study) by Hopi men. CRC screening rates (having had fecal occult blood test (FOBT) or colonoscopy) increased from 51.0% in 2012 [22] to 71.1% in 2018/2019 (*p* < 0.01). PCa screening rates (having had prostate specific antigen (PSA) test or digital rectal exam [DRE]) did not change (35.3% in 2012 and 37.4% in 2018/2019).

Among Hopi male participants who had CRC screening, colonoscopy (61.5%) was a more common screening method than FOBT (13.3%). About half of Hopi men (51.2%) reported having had a CRC screening (FOBT or colonoscopy) within the past 3 years (Table 2). For PCa screening (PSA or DRE), 60.7% of men self-reported having had PCa screening within the past 3 years.

Among Hopi male participants, having had a previous cancer screening increased their likelihood of having another type of cancer screening. Table 3 shows that men who had PCa screening were more likely to have a CRC screening (OR 5.33, 95% CI: 1.38–20.59). Similarly, Hopi male participants who had a CRC screening were more likely to have a PCa screening (OR 5.37, 95% CI: 1.36–21.17; Table 4). Cellular phone ownership was associated with an increased odds of PCa screening in the unadjusted model (OR 9.00, 95% CI: 1.11–73.07), but the association was not significant in the adjusted model.

## 4. Discussion

This paper describes the Community-Based Participatory Research process and activities. Through the participation of Hopi community members in this project, the research team successfully completed formative assessment for cancer screening behavior and mHealth approach for promotion of cancer screening in Hopi men.

The use of cellular phones was assessed as a method of communication between the Hopi Native Patient Navigator and Hopi men and as a method of cancer screening promotion for future projects or development of programs at H.O.P.I. Cancer Support Services. For some parts of the Hopi Reservation, internet access is either unavailable or unreliable. Web-based technologies (e.g., “secure patient portals”) commonly deployed in urban healthcare systems to support patient-provider communication are not practical on many reservations, including the Hopi Reservation. The 2012 Hopi Survey of Cancer and Chronic Disease als9 showed low home computer use in Hopi [47]. In this study, only three out of six Buddy Challenge Program participants reported that they owned a computer at home. mHealth that uses text messages to relay health messages may be an alternative approach for men living on a reservation. The use of mHealth technology has been successfully used in interventions on health promotion, cancer screening, and cancer care [34,48,49]. A clinic-based intervention study among American Indians/Alaska Natives in Alaska reported that text messaging increased CRC screening [35]. In this randomized controlled trial, the difference in the screening rates between intervention and control group was significant only in women, but not among men. Although Hopi men had low cellular phone use for healthcare and wellness and have experienced service issues regularly, cellular phone ownership increased odds of PCa screening in the unadjusted model. This could be potentially due to characteristics of cellular phone owners who had a better healthcare access than men who did not own cellular phone, and the association was not significant in the adjusted model. Nonetheless, smart phone ownership is expected to grow, and use of mHealth approach for health promotion and improvement of cancer screening participation should be further investigated in Hopi and other American Indian men.

This study found a high CRC screening rate and a low PCa screening rate among Hopi men. In 2010, the Hopi Tribe introduced a state-funded CRC screening promotion program, and this may have contributed to the increased CRC screening rate from 2012 to 2018–2019 [22]. On the other hand, the U.S. Preventative Services Task Force (USPSTF) does not recommend PSA screening, and for men with average risk, the USPSTF recommends individually based decision making after discussing benefits and harms of PCa screening with their healthcare provider [50]. Hopi men were concerned about the high number of PCa cases and deaths in their community, which was the main reason for including PSA screening in the program specifically for men. However, despite their concerns, the consistently low PCa screening rates in Hopi men may reflect the USPSTF’s recommendation against PCa screening. It also should be noted that differences in the study designs may have influenced the estimation of cancer screening rates. The 2012 Hopi Survey of Cancer and Chronic Disease was a population-based survey of randomly selected adult Hopi members living on the reservation, while this study used convenient sampling of Hopi men who were interested in attending ARS community survey sessions. Moreover, we previously noted that familiarity with cancer screening or exposure to the healthcare system through personal experience may increase cancer screening awareness and facilitate cancer screening behaviors among Hopi men [22]. In this study, previous cancer screening experience was the only factor associated with having CRC or PCa screening.

The research team encountered several challenges while conducting research in the Hopi community. Although Hopi and other tribes in Arizona have recognized the importance of developing health promotion projects focused on American Indian men’s health [51], recruitment of Hopi men was often challenging. As described previously, recruitment of study participants from racial/ethnic minority groups, especially American Indian/Alaska Native, for biomedical research is challenging [52,53,54], and enrollment for American Indian men in research studies is generally lower than for American Indian women [35,55,56]. In addition, Hopi is a small tribe, and residents are geographically spread across the reservation. Four ARS survey sessions also had very small participants (*n* < 3). Although Hopi community members have been involved in various research projects, many men were unfamiliar with research methods and activities. Some study participants anticipated a paper format survey rather than the ARS survey and decided not to participate after finding out how the survey was conducted. Sometimes men showed up to survey sessions after the participants had gone through more than half of the ARS survey questions, so they could not participate.

This study has other limitations. This was a pilot study, and convenience samples were used. Men from a limited area of the Hopi Reservation (mainly central areas on the reservation) may have participated more than men from peripheral areas of the reservation. The study has a small sample size. Therefore, the study findings may not be representative of the entire population of Hopi men, and the results may not be generalizable to men living in other areas on the Hopi Reservation. Similarly, the findings may not be generalizable to American Indian men from other tribal nations or those living in urban areas.

Despite challenges, this study demonstrated the Hopi team’s ability to recruit Hopi men to successfully complete formative assessment. First, this was a community-led initiative. After the successful development of a cancer prevention program for women, Hopi men requested a program specifically for men. Second, a Hopi male Native Patient Navigator who was bilingual in Hopi and English and was an active community member was instrumental in study participant recruitments and leading the ARS survey. The Native Patient Navigator conducted research activities as a part of community outreach promoting cancer screening program for men. Third, the team consulted regularly with community advisory committee members for recruitment strategies, and the committee members recommended raffle prizes for the ARS survey participants. Although the recruitment goal was not met, raffle prizes attracted many potential survey participants. Finally, the Hopi already had an established long-term partnership with university teams. Through this partnership, various community-based research projects were conducted leading to development of successful programs within the Hopi Department of Health and Human Services. The research team was able to utilize this established partnership to successfully initiate community engagements and research activities.

This Community-Based Participatory Research was successful in many other ways benefiting Hopi community members. First, by conducting and being involved in this project, the Hopi team, including the community advisory committee members, gained a better understanding of cancer, cancer screening methods and guidelines, cancer treatment, roles, and the responsibility of Native Patient Navigators. The Hopi team was able to understand how Native Patient Navigators could utilize a mHealth approach to promote cancer screening and communicate with Hopi men. The use of cellular phone for healthcare and wellness was not yet common in Hopi men. However, at-home computer and email usage were low suggesting that text messaging may be a better approach than emails. Moreover, through participation in the research activities, the study participants became aware of the support and services available from H.O.P.I. Cancer Support Services. Hopi men were very supportive of developing the Native Patient Navigator-led Hopi Men’s Cancer Program and reported interest in being involved in the process of program development. The research team presented study findings at the Hopi Tribal Council. The Hopi Tribal Council members recognized the importance of the roles that Hopi Native Patient Navigators played during and after the study period, and they expressed interests in supporting further addressing Hopi men’s needs. However, future research is necessary for successful and sustainable implementation of the Native Patient Navigator led cancer screening program using the mHealth approach for Hopi men.

## 5. Conclusions

The research team conducted the formative assessment of the mHealth approach and cancer screening in Hopi men living on the reservation with the engagement of Hopi community members. The mHealth approach may be appropriate for promotion of cancer screening in Hopi men, but more research is necessary to further assess if the Native Patient Navigator with mHealth approaches can increase cancer screening rates.

## Figures and Tables

**Figure 1 ijerph-19-06546-f001:**
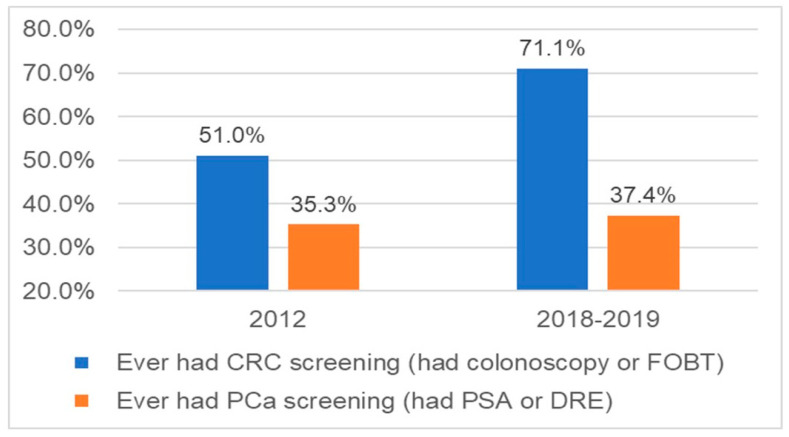
Cancer screening rates among Hopi men in 2012 (Hopi Survey of Cancer and Chronic Disease) and 2018–2019 (current study) survey. Abbreviations: Colorectal Cancer (CRC), Fecal Occult Blood Test (FOBT), Prostate Cancer (PCa), prostate specific antigen (PSA), and digital rectal exam (DRE).

**Table 1 ijerph-19-06546-t001:** Characteristics of Hopi men who participated the ARS surveys (*n* = 83).

Variables	*n* (%)
Reported Age	
50–64 years	54 (65.1%)
65–80 years	23 (27.7%)
81 years or older	2 (2.4%)
Missing	4 (4.8%)
Highest Reported Education	
Junior High	4 (4.8%)
High School but never completed	2 (2.4%)
High school graduate/GED	19 (22.9%)
Trade, technical or vocational school after High School	30 (36.1%)
Some college but no degree	23 (27.7%)
Bachelor degree or greater	5 (6.0%)
Married	30 (36.1%)
Employed full-time	27 (32.5%)
Household income	
Less than $10,000	18 (21.7%)
$10,000–$20,000	9 (10.8%)
$20,001–$30,000	11 (13.3%)
$30,001–$40,000	14 (16.9%)
≥$50,001	16 (19.3%)
Don’t know/Refused	15 (18.1%)
Own Cell phone	70 (84.3%)
Use cellular for texting	65 (78.3%)
Hopi Culture	
Primary language at home: Hopi	65 (78.3%)
Screening history	
Ever had colorectal cancer screening	59 (71.1%)
Ever had prostate cancer screening	31 (37.4%)
Cancer experience	
Has/had cancer	6 (7.2%)
Family members diagnosed with cancer	37 (44.5%)
Health Status	
Diabetes	36 (43.4%)
Hypertension	44 (53.0%)
High cholesterol	25 (30.1%)
Thinks his health is good, very good or excellent	59 (71.1%)
Hopi Health Care Center for primary care services	36 (43.4%)
Medicare/Medicaid, or AHCCCs health insurance	40 (48.2%)
Learn about cancer screening from	
Family members	16 (19.5%)
Friends	5 (6.1%)
Employers and coworkers	4 (4.9%)
Health care providers	11 (13.4%)
Community education events	6 (7.3%)
H.O.P.I. Cancer Support Services staff	28 (34.2%)
Media	6 (7.3%)
None of the above	6 (7.3%)
Last annual physical exam in 2018	43 (51.8%)
Looked for medical information in the past year using electronic devices	27 (32.5%)
Made medical appointment using electronic devices in the past years	24 (28.9%)
Sent/received text from health professionals	26 (31.3%)
Heath wellness apps on smartphone/tablet	16 (19.3%)
Received text from Hopi men’s health project	68 (81.4%)

Abbreviations: AHCCC, Arizona Health Care Cost Containment System, H.O.P.I., Hopi Office of Prevention and Intervention.

**Table 2 ijerph-19-06546-t002:** Type of cancer screening tests and years since the last screening test.

	FOBT	Colonoscopy	PSA	DRE
	11 (13.3%)	51 (61.5%)	14 (16.9%)	18 (21.7%)
Within the last year	8 (19.5%)		8 (28.6%)	
Within the last 3 years	13 (31.7%)		9 (32.1%)	
Within the last 5 years	12 (29.3%)		8 (28.6%)	
Within the last 10 years	5 (12.2%)		2 (7.1%)	
Longer than 10 years ago	3 (7.3%)		1 (3.6%)	
Missing	18		3	

Abbreviation: FOBT, Fecal Occult Blood Test; PSA, Prostate Specific Antigen, DRE, Digital Rectal Exam.

**Table 3 ijerph-19-06546-t003:** Identification of factors associated with having CRC screening for Hopi men with age ≥ 50.

	Unadjusted		Adjusted	
	OR (95% CI)	*p*	OR (95% CI)	*p*
Age < 65	0.24 (0.06–0.90)	0.03	0.35 (0.09–1.40)	0.14
Some college education	1.03 (0.38–2.80)	0.96		
Married	2.79 (0.92–8.50)	0.07		
Employed full-time	1.25 (0.44–3.50)	0.68		
Income < $40 K	1.39 (0.53–3.69)	0.51		
Own Cell phone	1.68 (0.49–5.77)	0.41		
Use cellular for texting	0.93 (0.29–2.98)	0.90		
Primary language at home: Hopi	0.42 (0.11–1.61)	0.20		
Ever had prostate screening	6.32 (1.70–23.51)	0.01	5.33 (1.38–20.59)	0.02
Cancer experience				
Has/had cancer	2.13 (0.24–19.25)	0.50		
Family history of cancer	0.93 (0.36–2.42)	0.88		
Diabetes	1.10 (0.42–2.88)	0.84		
Hypertension	3.13 (1.16–8.48)	0.02	2.56 (0.88–7.46)	0.09
High cholesterol	2.76 (0.83–9.16)	0.10		
Thinks his health is good, very good or excellent	1.76 (0.64–4.85)	0.27		
HHCC primary care services	0.54 (0.21–1.41)	0.21		
Medicare/Medicaid, or AHCCCs health insurance	0.56 (0.22–1.47)	0.24		
Last annual physical exam in 2018	1.11 (0.43–2.86)	0.83		
Looked for medical information in the past year using electronic devices	0.95 (0.35–2.61)	0.92		
Made medical appointment using electronic devices in the past years	0.57 (0.21–1.56)	0.27		
Sent/received text from health professionals	0.67 (0.25–1.83)	0.44		

Abbreviations: HHCC, Hopi Health Care Center; AHCCC, Arizona Health Care Cost Containment System. *p*-value determined by logistic regression.

**Table 4 ijerph-19-06546-t004:** Identification of factors associated with having PCa screening for Hopi men with age ≥ 50.

	Unadjusted		Adjusted	
	OR (95% CI)	*p*	OR (95% CI)	*p*
Age < 65	0.42 (0.16–1.09)	0.07		
Some college education	0.90 (0.35–2.31)	0.83		
Married	0.60 (0.23–1.57)	0.30		
Employed full-time	1.56 (0.61–3.99)	0.35		
Income < $40 K	1.05 (0.41–2.65)	0.92		
Own Cell phone	9.00 (1.11–73.07)	0.04	5.42 (0.60–3.89)	0.13
Use cellular for texting	2.49 (0.74–8.39)	0.14		
Primary language at home: Hopi	0.51 (0.18–1.47)	0.21		
Ever had CRC screening	6.32 (1.70–23.51)	0.01	5.37 (1.36–21.17)	0.02
Has/had cancer	3.70 (0.64–21.54)	0.15		
Family history of cancer	1.57 (0.64–3.86)	0.32		
Diabetes	0.91 (0.37–2.24)	0.84		
Hypertension	1.71 (0.69–4.23)	0.25		
High cholesterol	3.07 (1.16–8.11)	0.02	2.05 (0.70–6.02)	0.19
Thinks his health is good, very good or excellent	0.77 (0.29–2.04)	0.60		
HHCC primary care services	2.11 (0.85–5.21)	0.11		
Medicare/Medicaid, or AHCCCs health insurance	0.35 (0.14–0.89)	0.03	0.42 (0.15–1.19)	0.10
Last annual physical exam in 2018	0.99 (0.41–2.41)	0.98		
Looked for medical information in the past year using electronic devices	1.96 (0.77–5.02)	0.16		
Made medical appointment using electronic devices in the past years	2.11 (0.80–5.55)	0.13		
Sent/received text from health professionals	1.36 (0.53–3.51)	0.53		

Abbreviations: HHCC, Hopi Health Care Center; AHCCC, Arizona Health Care Cost Containment System. *p*-value determined by logistic regression.

## Data Availability

Due to a tribal regulation, the data obtained during this project will not be distributed freely. The collected data belong to Hopi Tribe. To access the data, a written request needs to be submitted to Hopi Tribe, and the request requires approvals from H.O.P.I. Cancer Support Services, Hopi Department of Health and Human Services, and Hopi Tribal Council.

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
