# Peer review of "Formative Assessment to Improve Cancer Screenings in American Indian Men: Native Patient Navigator and mHealth Texting"

_ijerph, 2022, doi:10.3390/ijerph19116546_

Round 1

Reviewer 1 Report

Formative assessment to improve cancer screenings in American Indian men: Native Patient Navigator and mHealth texting

Thank you for asking me to review this paper.

Overall:

The paper can be edited for grammar and clarity. There are instances of non-parallel language and grammar issues that can be easily fixed. For example,

Cancer screening rates among American Indian men remain low without programs specifically designed for men. This paper describes the Community-Based Participatory Research processes and activities and assess cancer screening behavior among Hopi men and appropriateness of mHealth approach for Hopi men’s promotion of cancer screenings.

Could be:

Cancer screening rates among American Indian men remain low without programs specifically designed for them. This paper describes the Community-Based Participatory Research processes and activities and assessment of (the paper is not assessing, as is written above) cancer screening behavior among Hopi men and appropriateness of an mHealth approach for Hopi men’s promotion of cancer screenings.

There are multiple other examples and the paper would be much improved if it were edited.

Introduction:

This section provides strong evidence of the need for programs such as the one described in this paper. It is well written and well cited with relevant and statistics. My assumption is that there are not any more current statistics available as many of the citations are dated.

Regarding this sentence: “As a result, female case managers have increased the rate of breast  cancer screening (have had a mammography within the past two years) from 26% in 1993 to 71% in 2012 [42, 43].” Please include who is in this group (age and is it the same as the men targeted for this project of residing on the Hopi reservation?). Are there any statistics that are more current?

It's nice to see that the need for this program arose from the community. The CAC has a strong and positive role in this research.

This sentence (lines 188-190) is confusing to me. Can you clarify what “educate research” means? The goals of the training were to prepare the first Hopi Native Patient Navigator to take on the role as a navigator and to educate research and Hopi community members on the roles and responsibilities of a navigator.

Materials and methods:

The section on Native Patient Navigator and Training does not seem very relevant to this paper. There’s a lot of detail here that does not seem necessary. You state that “appropriateness of the Hopi male Native Patient Navigator to provide support and service to Hopi men who are accessing cancer screenings and care was assessed as a part of the project.” But this does not seem a part of the results. Maybe I missed it?

Results:

Regarding lines 291-292: “Due to the small number of older Hopi men in the Buddy Challenge Program, the team lowered the eligibility criteria for age to 30 years and older upon a Northern Arizona University IRB approval.” The younger men may not be eligible to receive the screenings you were asking about. I didn’t see if they were not included in the analysis as their information may not be relevant.

Regarding lines 311-312: “Some study participants also enjoyed the interactive nature of ARS survey.” Was this a question on the ARS? How was this information gathered?

Table 1. Usually age is reported as a mean with a standard deviation.

I wasn’t sure where the data from Figure 1 came from. Were these questions on the ARS? Are these data comparable?

Table 3. Suggest that you add what is considered a significant finding. Delete “Tables may have a footer.”

It isn’t clear to me what additional information the interviews provided that the ARS didn’t. Is it the evaluation of text messaging? It seems like you received information on text messaging from the ARS participants as well. I’d like to see this more clearly described.

Discussion:

Regarding lines 380-382: “Although Hopi men had low cellular phone use for healthcare and wellness and have experienced service issues regularly, cellular phone ownership increased odds of PCa screening in unadjusted model.” Why do you think this is?

If the USPSTF does not recommend PSA screening, why is the article addressing it? I’m not sure what useful information this provides. This is a main comment of mine.

Regarding lines 426 and 427: “Despite challenges, this formative study demonstrated successful recruitment of Hopi men.” Your recruitment for the ARS was less than half of what was intended. Also, the age for eligibility for interviews was changed because of problems recruiting.

I’m not sure this would be categorized as successful recruitment.

Regarding lines 444-446: “The Hopi team was able to understand how Native Patient Navigators can utilize mHealth approach to promote cancer screening among Hopi men after the Hopi Men’s Health Program is further developed.” Only one of the Buddy Challenge participants preferred receiving health promotion messages by text messages. Is this statement based off of the “received text from Hopi men’s health project” question? It would be helpful to contextualize by stating what other data supports this in your findings.

It is promising that men in the community would like to see more programming.

Author Response

Thank you for asking me to review this paper.

RESPONSE: Thank you revising our manuscripts and providing useful comments.

Overall:

The paper can be edited for grammar and clarity. There are instances of non-parallel language and grammar issues that can be easily fixed. For example,

Cancer screening rates among American Indian men remain low without programs specifically designed for men. This paper describes the Community-Based Participatory Research processes and activities and assess cancer screening behavior among Hopi men and appropriateness of mHealth approach for Hopi men’s promotion of cancer screenings.

Could be:

Cancer screening rates among American Indian men remain low without programs specifically designed for men. This paper describes the Community-Based Participatory Research processes and activities and assessment of (the paper is not assessing, as is written above) cancer screening behavior among Hopi men and appropriateness of an mHealth approach for Hopi men’s promotion of cancer screenings.

There are multiple other examples and the paper would be much improved if it were edited.

RESPONSE: Thank you for your comments.  We went through entire paper, corrected any grammatical errors that we found, and clarified further.

Introduction:

This section provides strong evidence of the need for programs such as the one described in this paper. It is well written and well cited with relevant and statistics. My assumption is that there are not any more current statistics available as many of the citations are dated.

RESPONSE: Our challenge is that there are not many comprehensive research projects on cancer, especially cancer mortality, among American Indians and Alaska Natives in the United States. There are also many issues related to cancer statistics in this population (for example misclassification), so we tried to include only reliable cancer statistics in this paper.

Regarding this sentence: “As a result, female case managers have increased the rate of breast cancer screening (have had a mammography within the past two years) from 26% in 1993 to 71% in 2012 [42, 43].” Please include who is in this group (age and is it the same as the men targeted for this project of residing on the Hopi reservation?). Are there any statistics that are more current?

RESPONSE: The breast cancer study was for Hopi women aged 40 or older living in Hopi Reservation. This information was added the manuscript. The 2012 Hopi Survey of Cancer and Chronic Disease was the last population-based community survey conducted in Hopi and is the latest statistics available for Hopi women.  Our study is the latest statistics for Hopi men.

It's nice to see that the need for this program arose from the community. The CAC has a strong and positive role in this research.

RESPONSE: Thank you.

This sentence (lines 188-190) is confusing to me. Can you clarify what “educate research” means? The goals of the training were to prepare the first Hopi Native Patient Navigator to take on the role as a navigator and to educate research and Hopi community members on the roles and responsibilities of a navigator.

RESPONSE: We clarified the part of sentence.  Now, it says “to educate the research team and Hopi community members on …”

The section on Native Patient Navigator and Training does not seem very relevant to this paper. There’s a lot of detail here that does not seem necessary. You state that “appropriateness of the Hopi male Native Patient Navigator to provide support and service to Hopi men who are accessing cancer screenings and care was assessed as a part of the project.” But this does not seem a part of the results. Maybe I missed it?

RESPONSE: Our long-term goal was to develop a cancer screening program for Hopi men run by a Hopi male Patient Navigator. We agree that we did not have research findings relevant to patient navigator, but the patient navigator led the many research activities in Hopi community and promoted cancer screening program.  We discussed role of navigator in this study in the methods and discussion section.

Results:

Regarding lines 291-292: “Due to the small number of older Hopi men in the Buddy Challenge Program, the team lowered the eligibility criteria for age to 30 years and older upon a Northern Arizona University IRB approval.” The younger men may not be eligible to receive the screenings you were asking about. I didn’t see if they were not included in the analysis as their information may not be relevant.

RESPONSE: We interviewed the Buddy Challenge Program participants to assess cellular phone usage for a wellness program in older Hopi men and to gather information on appropriateness of the mHealth approach.  We did not collect information on cancer screening among them.  We tried to clarify the purpose of interviews in the methods and results section.

Regarding lines 311-312: “Some study participants also enjoyed the interactive nature of ARS survey.” Was this a question on the ARS? How was this information gathered?

RESPONSE: We expanded the description of ARS in the methods section. The questions and poll results (answers in percentage) were displayed on PowerPoint slides.

Table 1. Usually age is reported as a mean with a standard deviation.

RESPONSE: Because ARS uses multiple choice questions, study participants were not able provide their age in number.

I wasn’t sure where the data from Figure 1 came from. Were these questions on the ARS? Are these data comparable?

RESPONSE: We added information on survey conducted in 2012 (Hopi Survey of Cancer and Chronic Disease) and 2018/2019 (current study).

Table 3. Suggest that you add what is considered a significant finding. Delete “Tables may have a footer.”

RESPONSE: We considered p-value less than 0.05 statistically significant in all analysis. We added this sentence to the statistical analysis section of methods.  We deleted the “Tables May have a footer.”

It isn’t clear to me what additional information the interviews provided that the ARS didn’t. Is it the evaluation of text messaging? It seems like you received information on text messaging from the ARS participants as well. I’d like to see this more clearly described.

RESPONSE: The Buddy Challenge Program participants were men who chose to use cellular phone for wellness program.  They tend to be younger.  On the other hand, the ARS survey included anybody who were interested in participating in this study including those who did not have smartphone or who do not use text messaging. We had different sets of questions that we asked and reported here.

Discussion:

Regarding lines 380-382: “Although Hopi men had low cellular phone use for healthcare and wellness and have experienced service issues regularly, cellular phone ownership increased odds of PCa screening in unadjusted model.” Why do you think this is?

RESPONSE: This could be potentially due to characteristics of cellular phone owners who had a better healthcare access than men who did not own cellular phone and the association was not significant in the adjusted model. We added this to the discussion section.

If the USPSTF does not recommend PSA screening, why is the article addressing it? I’m not sure what useful information this provides. This is a main comment of mine.

RESPONSE: We understand the issues related to PSA screening.  However, Hopi men were concerned about high number of PCa cases and deaths in their community, which was the main reason for including PSA screening in the program specifically for men. We included this explanation in the revised manuscript.

Regarding lines 426 and 427: “Despite challenges, this formative study demonstrated successful recruitment of Hopi men.” Your recruitment for the ARS was less than half of what was intended. Also, the age for eligibility for interviews was changed because of problems recruiting.

I’m not sure this would be categorized as successful recruitment.

RESPONSE: We re-wrote the sentence.  Now, it says “Despite challenges, this study demonstrated Hopi team’s ability to recruit Hopi men to successfully complete formative assessment.”

Regarding lines 444-446: “The Hopi team was able to understand how Native Patient Navigators can utilize mHealth approach to promote cancer screening among Hopi men after the Hopi Men’s Health Program is further developed.” Only one of the Buddy Challenge participants preferred receiving health promotion messages by text messages. Is this statement based off of the “received text from Hopi men’s health project” question? It would be helpful to contextualize by stating what other data supports this in your findings.

RESPONSE: We still need assess if text messaging approach can be successfully implemented in Hopi community, but at-home computer and email usage were low in Hopi men, but cellular phone usage and ownership is expected to grow.

It is promising that men in the community would like to see more programming.

RESPONSE: Thank you.  We are looking forward to work with Hopi community and improve cancer screening among men.

Reviewer 2 Report

Dear Authors,

Thank you for the opportunity to review the manuscript titled, "Formative assessment to improve cancer screenings in American Indian men: Native Patient Navigator and mHealth texting." This work addresses an important issue for a vulnerable population. I recommend including narrative that addresses the following points in the Discussion section of the manuscript:

  1. Please describe how you determined an electronic means by which to conduct this work? Although similar work had been conducted in a female population, how did you determine that male Hopi participants would be open to an mHealth approach? Do you think this approach limited the findings? For example, were group or one-on-one interventions in person preferred?
  2. Please describe the next steps for this research.

Author Response

Thank you for the opportunity to review the manuscript titled, "Formative assessment to improve cancer screenings in American Indian men: Native Patient Navigator and mHealth texting." This work addresses an important issue for a vulnerable population. I recommend including narrative that addresses the following points in the Discussion section of the manuscript:

RESPONSE: Thank you for reviewing our manuscript and providing comments.

Please describe how you determined an electronic means by which to conduct this work? Although similar work had been conducted in a female population, how did you determine that male Hopi participants would be open to an mHealth approach? Do you think this approach limited the findings? For example, were group or one-on-one interventions in person preferred?

RESPONSE: We tried to justify use of text messaging for cancer screening promotion and communications between the patient navigator and Hopi men in the second paragraph of discussion.  This study investigated the use of text messaging and cellular phone for healthcare and wellness. We are working on another paper reporting if Hopi men are open to use mHeath approach. We noted that a future intervention or implementation research will be necessary to further assess how successfully implement the program combining text messaging and patient navigation. Effectiveness of group or one-on-one intervention can be assessed.

Please describe the next steps for this research.

Response: We added a sentence for the next step in the discussion. 

Reviewer 3 Report

This manuscript is well written. The CBPR methodology is very well thought out and executed. As a result not only are the findings relevant and innovative, the intervention will have sustainable implications on cancer prevention practices with this population. 

Author Response

This manuscript is well written. The CBPR methodology is very well thought out and executed. As a result, not only are the findings relevant and innovative, the intervention will have sustainable implications on cancer prevention practices with this population.

Response: Thank you for reviewing our manuscript.